# Bayesian Multi-Task Transfer Learning for Soft Prompt Tuning

**Haeju Lee**[1*]      **Minchan Jeong**[1*]      **Se-Young Yun**[1]      **Kee-Eung Kim**[1]

[1]Kim Jaechul Graduate School of AI, KAIST

{lhg912, mcjeong, yunseyoung, kekim}@kaist.ac.kr

## Abstract

Prompt tuning, in which prompts are optimized to adapt large-scale pre-trained language models to downstream tasks instead of fine-tuning the full model parameters, has been shown to be particularly effective when the prompts are trained in the multi-task transfer learning setting. These methods generally involve individually training prompts for each source task and then aggregating them to provide the initialization of the prompt for the target task. However, this approach critically ignores the fact that some of the source tasks could be negatively or positively interfering with each other. We argue that when we extract knowledge from source tasks via training source prompts, we need to consider this correlation among source tasks for better transfer to target tasks. To this end, we propose a Bayesian approach where we work with the posterior distribution of prompts across source tasks. We obtain representative source prompts corresponding to the samples from the posterior utilizing Stein Variational Gradient Descent, which are then aggregated to constitute the initial target prompt. We show extensive experimental results on the standard benchmark NLP tasks, where our Bayesian multi-task transfer learning approach outperforms the state-of-the-art methods in many settings. Furthermore, our approach requires no auxiliary models other than the prompt itself, achieving high degree of parameter-efficiency.[1]

## 1 Introduction

Large-scale pre-trained language models (PLMs) have been recently fine-tuned for various NLP tasks (Devlin et al., 2019; Raffel et al., 2020a). Due to the computational challenges of training the extensive parameters in PLMs, there is a growing focus on methods that efficiently tune fewer parameters (Houlsby et al., 2019; Ben Zaken et al., 2022).

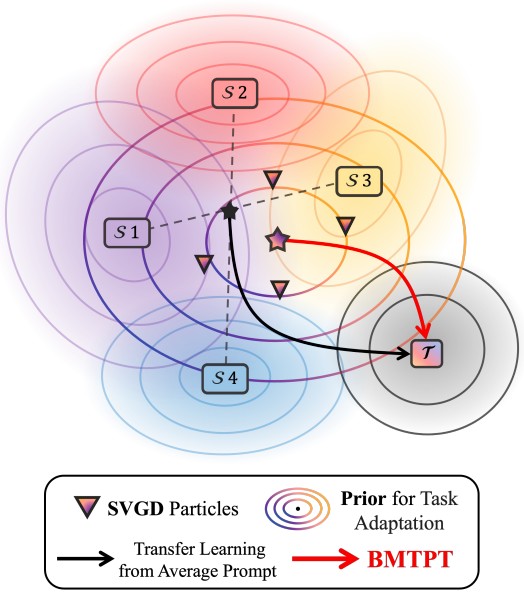

Figure 1: Two key steps for Bayesian Multi-Task Prompt Tuning (BMTPT) are illustrated. First, we merge the posterior distributions of each source task to form a global posterior distribution. This distribution is approximated using Stein Variational Gradient Descent (SVGD), a particle-based variational inference method. Finally, we adapt to the target task by using the derived posterior from the source tasks as a prior.

One of the promising approaches is prompt tuning (PT, Lester et al. 2021), where a few adaptable vectors are added as prompts to the input of the downstream task (Lester et al., 2021; Li and Liang, 2021). PT freezes the PLM model parameters and limits the learning to prompts, yet it achieves impressive performance. However, it is still challenging to achieve the same level of performance as the full fine-tuning, as well as to mitigate sensitivity to initialization (Zhong et al., 2022).

To address these challenges, recent works (Wang et al., 2023; Asai et al., 2022; Vu et al., 2022) proposed to adopt multi-task transfer learning approach, where the prompt is trained from multiple source tasks to be applied to the target task. Specifically, they train real-valued vectors for prompts

---

[*]These authors contributed equally to this work

[1]Code: https://github.com/heyzude/BMTPT

(i.e. soft prompts) on source tasks and use them as the initialization of prompt for the target task. However, it is unclear whether aggregating such individually-trained prompts provides a reliable initialization point and fully harnesses the benefits of multi-task transfer learning.

In this paper, we propose Bayesian Multi-Task Prompt Tuning (BMTPT) as a practical yet effective solution to this challenge. Unlike traditional methods of prompt tuning grounded in transfer learning, our approach engages with the posterior distribution of prompts across a multitude of source tasks. For the transference of knowledge gleaned from source tasks, we utilize the source prompts' posterior distribution as the prior for the designated target task. This Bayesian method of transfer learning augments the conventional transfer learning framework, which primarily learns the initialization point of the target prompt from the source tasks. Specifically, BMTPT employs Stein Variational Gradient Descent (SVGD, Liu and Wang 2016), a particle-based Variational Inference (VI) method, to approximate the source prompts' posterior distribution. Further elaboration on this method is provided in Section 2.2.

We validate our approach through experiments on 21 datasets across diverse NLP tasks and output formats. The experimental results demonstrate that BMTPT achieves comparable or superior performance to strong state-of-the-art parameter-efficient fine-tuning methods (Asai et al., 2022; Wang et al., 2023) as well as full fine-tuning, while utilizing a very small number of parameters and requiring no auxiliary models other than the prompt itself.

## 2 Background

### 2.1 Transfer Learning for Prompt Tuning

Fine-tuning entire models for downstream NLP tasks, particularly with a Large Language Model (LLM), can be expensive in terms of training costs. Therefore, parameter-efficient tuning focuses on limiting the updates to a small set of parameters. Various approaches have been proposed, such as Adapter (Houlsby et al., 2019) and its variants (Karimi Mahabadi et al., 2021a; Hu et al., 2022) that involve inserting trainable layers, and BitFit (Ben Zaken et al., 2022) that only trains bias weights while keeping other weights intact.

Recently, there has been growing interest in prompt tuning (PT). This approach involves updating only the 'soft prompt', a set of continuous

vectors that are prepended to the input. We can formally describe PT as follows: consider an input sequence $x$, and a soft prompt $\theta \in \mathbb{R}^{l \times d}$ with length $l$ and dimension $d$, which matches the language model's (LM) embedding dimension. The soft prompt is prepended to the sequence $x$ and then processed by the LM, resulting in the prediction of the target sequence $y$.

Our work aligns closely with recent efforts to transfer soft prompts from source tasks in order to initialize prompts for target tasks. For instance, SPoT (Vu et al., 2022) retrieves a source task prompt based on similarity to initialize the target task prompt, while ATTEMPT (Asai et al., 2022) employs an attention mechanism to initialize the prompt for the target task using information from the source prompts[2]. The most recent method for prompt tuning transfer, MPT (Wang et al., 2023), decomposes source prompts into shared and task-specific parts to reduce interference between tasks during aggregation.

However, these strategies may not fully address the inherent heterogeneity within source task distributions. They can falter, especially when attempting to aggregate prompts that have been trained across various tasks. Particularly, these issues persist even when each source task's posterior distribution follows a Gaussian distribution. Further discussion on this subject can be found in Appendix A.

Hence, an integrated approach regarding source task distributions may prove advantageous if a representative knowledge set can be constituted for transfer to the target task. This paper takes a Bayesian approach to transferring prompts from source tasks to target tasks. Instead of learning prompts individually then aggregating them, we use the full posterior distribution of prompts *across* the source tasks. Since this is intractable, we approximate the posterior via sampling, and leverage these samples for training the prompt for the target task, which corresponds to setting the posterior as the prior of the target prompt.

### 2.2 Particle Based VI and SVGD

Variational Inference (VI) is a widely used approach in machine learning for distribution approximation, notable in Bayesian Neural Networks (BNNs) (Blundell et al., 2015; Graves, 2011). Despite its computational simplicity, it often restricts

---

[2]Although ATTEMPT includes a randomly initialized target prompt in the attentional mixture, the argument in this section still applies.

the family of distributions, a limitation that is less present in methods like MCMC (Gilks et al., 1995; Doucet et al., 2001; Robert and Casella, 2004).

Particle-based VI methods provide an alternative approach by drawing upon the strengths of both VI and MCMC. Unlike traditional VI methods, particle-based VI does not restrict itself to a specific family of distributions. This flexibility allows it to approximate a wider range of complex and diverse distributions (Liu and Wang, 2016; Zhang et al., 2020; Naesseth et al., 2018). However, its theoretical guarantees are not yet fully understood. Assumptions often made, such as the presence of infinite particles or adherence to simple distributions like Gaussian, may not hold in practical scenarios (Naesseth et al., 2018; Salim et al., 2022; Sun et al., 2022; Liu et al., 2023).

Stein Variational Gradient Descent (SVGD, Liu and Wang 2016) is a significant advancement in particle-based VI. SVGD applies a transformation to particles to make them more representative of the target distribution through an iterative process. Specifically, for let particles tend to position themselves as though they were samples drawn from the distribution $p$, The update rule of SVGD is described as follows:

$$\boldsymbol{\theta}_i \leftarrow \boldsymbol{\theta}_i + \alpha \boldsymbol{\phi}_p^*(\boldsymbol{\theta}_i), \text{where } \boldsymbol{\phi}_p^*(\boldsymbol{\theta}_i) \text{ is}$$

$$\frac{1}{M} \sum_{j=1}^{M} k(\boldsymbol{\theta}_j, \boldsymbol{\theta}_i) \nabla_{\boldsymbol{\theta}_j} \log p(\boldsymbol{\theta}_j) + \nabla_{\boldsymbol{\theta}_j} k(\boldsymbol{\theta}_j, \boldsymbol{\theta}_i) ,$$

where $k(\cdot, \cdot)$ is the positive definite kernel function like RBF.

Despite its merits, SVGD can face mode collapse (Chen and Ghattas, 2020; Liu et al., 2022). One workaround, Damped SVGD (Ba et al., 2022), mitigates this by adjusting the deterministic bias in the update rule, which is used in our work. For a more thorough mathematical explanation, kernel details, and information about damped SVGD, we direct readers to Appendix B.

## 3   Problem Setting

In this section, we formally introduce core elements, symbols, and problem statements that form the basis of our approach. We denote the trainable parameter of the soft prompt as $\boldsymbol{\theta} \in \mathbb{R}^{l \times d}$, characterized by its length $l$ and the dimension $d$ of the Language Model (LM). For clarity, we use $\boldsymbol{\theta}^{\mathcal{S}}$ and $\boldsymbol{\theta}^{\mathcal{T}}$ to denote the soft prompts for source tasks and target task(s), respectively. This implies that the

soft prompt $\boldsymbol{\theta}$ is prepended to the sequence $\boldsymbol{x}$, prior to its processing by the LM. The underlying objective is to predict the target sequence $\boldsymbol{y}$. We denote the dataset for the $k$-th source as $\mathcal{D}_k^{\mathcal{S}}$, and define $\mathcal{D}^{\mathcal{S}} = \bigcup_{k=1}^{K} \mathcal{D}_k^{\mathcal{S}}$. The target task is represented as $\mathcal{D}^{\mathcal{T}}$ during task adaptation. Thus, the $i$-th instance in the $\mathcal{D}_k^{\mathcal{S}}$ dataset will be represented as $(\boldsymbol{x}_i^k, \boldsymbol{y}_i^k)$. Note that the log-likelihood $\log p(\mathcal{D}_k^{\mathcal{S}} | \boldsymbol{\theta}^{\mathcal{S}})$ for the $k$-th source task can be represented as follows:

$$\log p(\mathcal{D}_k^{\mathcal{S}} | \boldsymbol{\theta}^{\mathcal{S}}) = \sum_{i=1}^{|\mathcal{D}_k^{\mathcal{S}}|} \log p_{\text{LM}}(\boldsymbol{y}_i^k | [\boldsymbol{\theta}^{\mathcal{S}}; \boldsymbol{x}_i^k]) .$$

In this formulation, $p_{\text{LM}}$ denotes the likelihood determined by the LM and the corresponding criterion.

Next we state our Bayesian objective, aiming to optimize the target task prompt using the posterior of source prompts for a transfer learning scheme.

**Problem Statement.** *The objective is to maximize the posterior probability of the target prompt $\boldsymbol{\theta}^{\mathcal{T}}$, as expressed by the following equation:*

$$\underset{\boldsymbol{\theta}^{\mathcal{T}}}{\arg\max} \; p(\mathcal{D}^{\mathcal{T}} | \boldsymbol{\theta}^{\mathcal{T}}) \, p(\boldsymbol{\theta}^{\mathcal{T}} | \mathcal{D}^{\mathcal{S}}) , \qquad (1)$$

*where $p(\mathcal{D}^{\mathcal{T}} | \boldsymbol{\theta}^{\mathcal{T}})$ is the likelihood and $p(\boldsymbol{\theta}^{\mathcal{T}} | \mathcal{D}^{\mathcal{S}})$ is the prior that is learned from the source tasks in prior to the target task adaptation:*

$$p(\boldsymbol{\theta}^{\mathcal{T}} | \mathcal{D}^{\mathcal{S}}) = \int_{\boldsymbol{\theta}^{\mathcal{S}}} p(\boldsymbol{\theta}^{\mathcal{T}} | \boldsymbol{\theta}^{\mathcal{S}}) \, p(\boldsymbol{\theta}^{\mathcal{S}} | \mathcal{D}^{\mathcal{S}}) d\boldsymbol{\theta}^{\mathcal{S}} . \quad (2)$$

*In this context, the prior distribution $p(\boldsymbol{\theta}^{\mathcal{T}} | \mathcal{D}^{\mathcal{S}})$ serves as a guide for the target task adaptation. We model $p(\boldsymbol{\theta}^{\mathcal{T}} | \boldsymbol{\theta}^{\mathcal{S}})$ as the multivariate Gaussian with mean $\boldsymbol{\theta}^{\mathcal{S}}$, since without any information on the target task, it is natural to have $\boldsymbol{\theta}^{\mathcal{T}} = \boldsymbol{\theta}^{\mathcal{S}}$.*

This problem formulation provides a general framework subsuming conventional transfer learning method for prompt tuning. For example, we could approximate the above integral in Eq. (2) defining the prior on $\boldsymbol{\theta}^{\mathcal{T}}$ using a prompt trained from source tasks $\boldsymbol{\theta}^{\mathcal{S}*}$, i.e. $p(\boldsymbol{\theta}^{\mathcal{S}} | \mathcal{D}^{\mathcal{S}}) = \delta_{\boldsymbol{\theta}^{\mathcal{S}*}}(\boldsymbol{\theta}^{\mathcal{S}})$, which would be roughly equivalent to the conventional transfer learning setting where the source prompt serves as the initialization of the target prompt.

Assuming an uninformative prior for the source prompt $\boldsymbol{\theta}^{\mathcal{S}}$ (e.g. uniform distribution) as well as independent selection of source tasks, the posterior distribution $p(\boldsymbol{\theta}^{\mathcal{S}} | \mathcal{D}^{\mathcal{S}})$ for source tasks is formulated as the product of the posteriors of each task.

**Remark.** *Assuming the uniform prior for $\boldsymbol{\theta}^{\mathcal{S}}$ and independent selection of source tasks, the global posterior $p(\boldsymbol{\theta}^{\mathcal{S}} \mid \mathcal{D}^{\mathcal{S}})$ is proportional to the product of posteriors:*

$$p\big(\boldsymbol{\theta}^{\mathcal{S}} \,\big|\, \mathcal{D}^{\mathcal{S}} = \textstyle\bigcup_{k=1}^{K} \mathcal{D}_k^{\mathcal{S}}\big) \propto \prod_{k=1}^{K} p(\boldsymbol{\theta}^{\mathcal{S}} \,|\, \mathcal{D}_k^{\mathcal{S}}).$$

## 4 Approach

Instead of optimizing individual prompts for each source task in isolation, our method primarily revolves around learning the posterior distribution of source prompts across all source tasks. This approach assigns a larger probability mass to those prompts capable of addressing a greater number of source tasks, thereby potentially becoming more suitable candidate prompts for the target task as well. We implement this concept by using particles to approximate the posterior distribution. The following subsections provide a detailed explanation of this methodology.

### 4.1 Main Strategy

The optimization of the target task prompt is modeled as a MAP inference in Eq. (1) using $p(\boldsymbol{\theta}^{\mathcal{T}} | \mathcal{D}^{\mathcal{S}})$ as the prior. We approximate this with $M$ particles $\{\boldsymbol{\theta}_i^{\mathcal{S}}\}_{i=1}^{M}$ (each particle corresponds to a soft prompt) drawn from $p(\,\cdot\,|\mathcal{D}^{\mathcal{S}})$ using SVGD:

$$p(\boldsymbol{\theta}^{\mathcal{T}}|\mathcal{D}^{\mathcal{S}}) = \mathbb{E}\left[p(\boldsymbol{\theta}^{\mathcal{T}}|\boldsymbol{\theta}^{\mathcal{S}}) \,\Big|\, \boldsymbol{\theta}^{\mathcal{S}} \sim p(\,\cdot\,|\mathcal{D}_s)\right]$$
$$\overset{\text{Monte-Carlo}}{\underset{\text{Sampling}}{\simeq}} \frac{1}{M} \sum_{i=1}^{M} p(\boldsymbol{\theta}^{\mathcal{T}}|\boldsymbol{\theta}_i^{\mathcal{S}}). \quad (3)$$

For task adaptation, i.e. obtaining the prompt for the target task, the objective Eq. (1) is achieved based on the approximation provided by Eq. (3):

$$\underset{\boldsymbol{\theta}^{\mathcal{T}}}{\operatorname{argmin}} -\log p(\mathcal{D}^{\mathcal{T}} \,|\, \boldsymbol{\theta}^{\mathcal{T}}) - \log p(\boldsymbol{\theta}^{\mathcal{T}} \,|\, \mathcal{D}^{\mathcal{S}})$$
$$\simeq \underset{\boldsymbol{\theta}^{\mathcal{T}}}{\operatorname{argmin}} -\log p(\mathcal{D}^{\mathcal{T}} | \boldsymbol{\theta}^{\mathcal{T}}) \underbrace{- \log \frac{1}{M} \sum_{i=1}^{M} p(\boldsymbol{\theta}^{\mathcal{T}}|\boldsymbol{\theta}_i^{\mathcal{S}})}_{=: \, J(\boldsymbol{\theta}^{\mathcal{T}})}$$
$$(4)$$

The pseudo-code of our BMTPT algorithm is shown in Algorithm 1.

For practical purposes, we can minimize the second term of the objective $J(\boldsymbol{\theta}^{\mathcal{T}})$ by applying Jensen's inequality, as demonstrated below:

$$-\log \frac{1}{M} \sum_{i=1}^{M} p(\boldsymbol{\theta}^{\mathcal{T}}|\boldsymbol{\theta}_i^{\mathcal{S}}) \leq -\frac{1}{M} \sum_{i=1}^{M} \log p(\boldsymbol{\theta}^{\mathcal{T}}|\boldsymbol{\theta}_i^{\mathcal{S}})$$
$$= \frac{1}{2\sigma^2} \left\| \boldsymbol{\theta}^{\mathcal{T}} - \frac{1}{M}\sum_{i=1}^{M} \boldsymbol{\theta}_i^{\mathcal{S}} \right\|^2 + C$$
$$(5)$$

---

**Algorithm 1** Bayesian Multi-Task Prompt Tuning

> **Input:**
> $\mathcal{D}^{\mathcal{S}}, \mathcal{D}^{\mathcal{T}}$ : source tasks and target task
> $\boldsymbol{\Theta}_0 = \{\boldsymbol{\theta}_{0,i}\}_{i=1}^{M}$ : initialized particle set

**Source Posterior Learning:**
**for** $t \leftarrow 0$ to $T-1$ **do**
    $\boldsymbol{\Theta}_{t+1} \leftarrow \boldsymbol{\Theta}_t + \alpha \boldsymbol{\phi}_{p(\cdot|\mathcal{D}^{\mathcal{S}})}^{*}(\boldsymbol{\Theta}_t)$
    (SVGD iteration; Section 2.2)
**end for**
Store $\boldsymbol{\theta}_i^{\mathcal{S}} \leftarrow \boldsymbol{\theta}_{T,i}$ for all $i \in [M]$

**Target Task Adaptation:**
$J(\boldsymbol{\theta}^{\mathcal{T}}) = -\log p(\mathcal{D}^{\mathcal{T}}|\boldsymbol{\theta}^{\mathcal{T}}) - \log \frac{1}{M}\sum_{i=1}^{M} p(\boldsymbol{\theta}^{\mathcal{T}}|\boldsymbol{\theta}_i^{\mathcal{S}})$
$\boldsymbol{\theta}^{\mathcal{T}*} \leftarrow \operatorname{argmin}_{\boldsymbol{\theta}^{\mathcal{T}}} J(\boldsymbol{\theta}^{\mathcal{T}})$

> **Output:**
> $\boldsymbol{\theta}^{\mathcal{T}*}$ : trained weight for the target task

---

where $\sigma$ and $C$ are constants arising from the multivariate isotropic Gaussian assumption of $p(\boldsymbol{\theta}^{\mathcal{T}}|\boldsymbol{\theta}^{\mathcal{S}})$. Combining Eq. (4) and Eq. (5), the final loss for target adaptation is therefore:

$$\underset{\boldsymbol{\theta}^{\mathcal{T}}}{\operatorname{argmin}} \left[ -\log p(\mathcal{D}^{\mathcal{T}}|\boldsymbol{\theta}^{\mathcal{T}}) + \frac{1}{2\sigma^2} \left\| \boldsymbol{\theta}^{\mathcal{T}} - \bar{\boldsymbol{\theta}}^{\mathcal{S}} \right\|^2 \right]$$
$$(6)$$

where $\bar{\boldsymbol{\theta}}^{\mathcal{S}} = \frac{1}{M}\sum_{i=1}^{M} \boldsymbol{\theta}_i^{\mathcal{S}}$. This objective suggests that, during target adaptation, we can initialize $\boldsymbol{\theta}^{\mathcal{T}}$ with the average value of the optimized particles $\boldsymbol{\theta}^{\mathcal{T}} \leftarrow \bar{\boldsymbol{\theta}}^{\mathcal{S}}$.

### 4.2 Additional Strategies

#### 4.2.1 Source Task Sampling

As transfer learning prepares for unknown arbitrary target tasks, usually it is considered preferable that various source tasks are learned. However, if the number of source tasks $K$ increases, we have to calculate the training losses of all source tasks. Therefore it is necessary to alleviate the bottleneck coming from a large number of source tasks. To this end, we use an approximate posterior distribution instead of the true global posterior distribution. Specifically, during each source posterior learning iteration, we uniformly sample $\kappa$ tasks from the $K$ source tasks ($\kappa < K$) without replacement and constitute a batch with the data entries from that $\kappa$ tasks.

### 4.2.2 Composition of $\theta^{\mathcal{T}}$ and Multi-target Task Adaptation

At the start of the target adaptation, we compose $\theta^{\mathcal{T}}$ with element-wise multiplication of a full-rank matrix which is initialized with $\bar{\theta}^{\mathcal{S}}$ and a low-rank matrix whose elements are all 1, where both matrices are learnable and have the shape of $(l, d)$. The low-rank matrix is made by $\boldsymbol{ab}^{\top}$ where $\boldsymbol{a} = \boldsymbol{1}^l$ and $\boldsymbol{b} = \boldsymbol{1}^d$ and both $\boldsymbol{a}$, $\boldsymbol{b}$ are trainable components. Importantly, during target adaptation, we adopt a two-speed learning rate scheme for the full-rank and low-rank matrices by setting a higher learning rate for the low-rank matrix (Ponti et al., 2022; Asai et al., 2022; Wang et al., 2023). This facilitates multi-target task adaptation, by employing multiple low-rank matrices to assign each low-rank matrix to each target task, while sharing the full-rank matrix among all target tasks. In doing so, the full-rank matrix captures the shared knowledge across tasks, while the respective low-rank matrices capture the task-specific knowledge (Wang et al., 2023). We also apply this scheme to single-target adaptation, as we empirically observed that the use of two-speed learning rate promotes faster performance convergence.

### 4.3 Training Process

#### 4.3.1 Source Task Posterior Approximation

Unlike previous transfer learning methods in prompt tuning that individually train prompts for each source task, we approximate the global posterior distribution of source tasks, by employing $M$ particles. Here, a particle corresponds to one instance of soft prompt. Each particle is initialized with randomly sampled tokens, following Lester et al. (2021). We pack a batch as the following: each particle $\theta_i^{\mathcal{S}}$ (which is an instantiation of soft prompt, and $1 \leq i \leq M$) is prepended to input texts from $K$ source tasks, forming a batch of size $M \cdot K$. It is worth noting that the $\log p(\cdot)$ in the SVGD update rule can be interpreted as the minus of the cross-entropy loss of the language model when given the input with the particle (prompt) prepended. Also, we employ a limited number of SVGD particles, usually $M \leq 10$. We perform 100K SVGD updates to sample $\theta^{\mathcal{S}}$.

#### 4.3.2 Target Task Adaptation

With the initialized $\theta^{\mathcal{T}}$, we start target task adaptation. The loss for the adaptation process is Eq. (6), which is the combination of Maximum Likelihood Estimation (MLE) loss with respect to $\mathcal{D}^{\mathcal{T}}$ and minus of the average of log priors.

### 4.4 Efficiency of BMTPT

Recent prompt tuning transfer methods primarily focus on measuring the efficiency during target adaptation, overlooking the need to evaluate the efficiency of source task training phase, which is helpful for identifying potential bottlenecks. We highlight the efficiency of BMTPT in comparison to the most recent prompt tuning transfer methods, ATTEMPT (Asai et al., 2022) and MPT (Wang et al., 2023), in both source and target stage. It is noteworthy that both methods require additional neural networks beyond soft prompts during either source task training or target adaptation: MPT involves a teacher network that is of the same size as the LM backbone as it uses distillation during source task training, and ATTEMPT involves the training of an attention module during target adaptation.

BMTPT, on the other hand, proves to be efficient in both the source posterior learning and target adaptation stages, when evaluated under criteria of computational and space complexity. The additional intricacies that BMTPT introduces, compared to vanilla prompt tuning, are the use of SVGD during source posterior learning and the computation of regularization terms derived from the prior during target adaptation (Eq. (6)). In terms of computational complexity, given that the SVGD step used in BMTPT primarily involves computing RBF kernel values among a limited number of particles, the computational cost is minimal. Likewise, the regularization calculation during target adaptation is also negligible. On the aspect of space complexity, BMTPT continues to exhibit efficiency. During source posterior learning, as BMTPT accompanies SVGD particles only, the memory space that BMTPT requires is occupied by the backbone LM parameters and the SVGD particles which are comprised of $M \cdot l \cdot d$ trainable parameters. Since we employ a small number of particles, the memory consumption by SVGD particles is almost negligible. During target adaptation, as we compose one target task prompt with shared matrix (full-rank) and task-specific matrix (low-rank), BMTPT requires $(l \cdot d)/N + (l + d)$ trainable parameters per one target task, when we adapt on $N$ target tasks. This makes BMTPT train only 0.035% parameters compared to full fine-tuning. For a detailed

analysis, we direct the reader to Appendix C.

## 4.5 Contrasts and Contributions

### 4.5.1 Constrast with Conventional Multi-Task Learning

Both BMTPT and traditional multi-task learning algorithms have a common point in that they utilize multi-source data. However, BMTPT uses multi-source data to find a posterior distribution across the multi-source data and transfer the posterior to target domain, under the Bayesian perspective. Traditional multi-task learning methods, on the other hand, optimize network parameters with respect to MLE objectives in general.

### 4.5.2 Distinction from MPT

BMTPT constructs soft prompt using full-rank and low-rank matrices during the target adaptation stage, similar to MPT (Wang et al., 2023). However, the methodologies diverge in their application and intent. MPT separates shared (full-rank) and task-specific (low-rank) components during source training with the underlying intuition that discovering common patterns among various source tasks can promote efficient transfer. After source training, MPT re-uses the shared component and averages task-specific components to initialize full-rank and low-rank matrices, then applies element-wise multiplication of the full and low-rank matrices to form a target prompt. BMTPT, on the other hand, does not use the decomposition at source posterior learning. We employ SVGD particles which are instantiations of full-rank prompts, to learn the source posterior. Then, at the beginning of target adaptation, we prepare a full-rank matrix and low-rank matrix to form a target prompt. The full-rank matrix is initialized with the average of SVGD particles and the low-rank matrix is initialized with $\mathbf{1}^{l \times d}$ (see Section 4.2.2). Notably, the intention behind this prompt decomposition is different from that of MPT; our aim is only to facilitate multi target task adaptation.

### 4.5.3 Unique Motivation behind BMTPT

At its core, BMTPT aims to focus on the essence of transfer learning by enhancing it via transferring a useful distribution as a prior for target adaptation. On the other hand, existing prompt transfer methods such as SPoT, ATTEMPT, and MPT tend to rely on the transferability between individual NLP tasks (e.g., SQuAD is more helpful for solving MRPC than SST-2). The efficacy of this unique motivation is demonstrated by the experimental results in Section 6.

## 5 Experiment

### 5.1 Datasets and Tasks

As in previous works (Asai et al., 2022; Wang et al., 2023), We use a set of 6 extensive datasets as source tasks and assess the performance of our algorithm on a range of 21 distinct target tasks, encompassing entailment, paraphrase detection, sentiment analysis, question answering (QA), and commonsense reasoning.

**Source Tasks** During source posterior learning, we use the following datasets from GLUE (Wang et al., 2019b), SuperGLUE (Wang et al., 2019a), and MRQA 2019 shared task (MRQA; Fisch et al. 2019), comprising over 100,000 annotations in total. Specifically, we utilize 6 source tasks, MNLI (Williams et al., 2018), QNLI (Demszky et al., 2018), QQP (Wang et al., 2019b) and SST-2 (Socher et al., 2013) from GLUE, SQuAD (Rajpurkar et al., 2016) from MRQA, and ReCoRD (Zhang et al., 2018) from SuperGLUE.

**Target Tasks** For target adaptation, we test our algorithm with 21 datasets from four benchmarks: MNLI, QQP, QNLI, SST-2, RTE (Giampiccolo et al., 2007), CoLA (Warstadt et al., 2019), STS-B (Cer et al., 2017) and MRPC (Dolan and Brockett, 2005) from GLUE; BoolQ (Clark et al., 2019), CB (de Marneffe et al., 2019), MultiRC (Khashabi et al., 2018), WiC (Pilehvar and Camacho-Collados, 2019) and WSC (Levesque et al., 2012) from SuperGLUE; Natural Questions (NQ; Kwiatkowski et al. 2019), HotpotQA (HQ; Yang et al. 2018), NewsQA (News; Trischler et al. 2017) and SearchQA (SQA; Dunn et al. 2017) from MRQA; WinoGrande (Sakaguchi et al., 2020), Yelp-2 (Zhang et al., 2015), SciTail (Khot et al., 2018) and PAWS-Wiki (Zhang et al., 2019) from the "Others" benchmark in Asai et al. (2022). We direct readers to Appendix D for the performance and analysis on MRQA and "Others" benchmarks.

### 5.2 Implementation Details and Baselines

**Implementation Details** Throughout the experiments, we use T5-base as the base LM for BMTPT and all of the baselines, and we use prompt of length 100. Unless specified differently, we

Table 1 data with headers:

| Method | # Params | GLUE | | | | | | | | | SuperGLUE | | | | | |
|---|---|---|---|---|---|---|---|---|---|---|---|---|---|---|---|---|
| | | MNLI | QQP | QNLI | SST-2 | STS-B | MRPC | RTE | CoLA | Avg. | Multirc | BoolQ | WiC | WSC | CB | Avg. |
| Fine-tuning | \|LM\| | 86.8 | 91.6 | 93.0 | 94.6 | 89.7 | 90.2 | 71.9 | 61.8 | 84.9 | 72.8 | 81.1 | 70.2 | 59.6 | 85.7 | 73.9 |
| Adapters | 1.9M | **86.5** | 90.2 | 93.2 | 93.8 | 90.7 | 85.3 | 71.9 | 64.0 | 84.5 | **75.9** | **82.5** | 67.1 | **67.3** | 85.7 | **75.7** |
| BitFit | 280K | 85.3 | 90.1 | 93.0 | 94.2 | **90.9** | 86.8 | 67.6 | 58.2 | 83.3 | 74.5 | 79.6 | **70.0** | 59.6 | 78.6 | 72.5 |
| PT | 76.8K | 81.3 | 89.7 | 92.8 | 90.9 | 89.5 | 68.1 | 54.7 | 10.6 | 72.2 | 58.7 | 61.7 | 48.9 | 51.9 | 67.9 | 57.8 |
| Vanilla transfer PT | 76.8K | 85.8 | 86.9 | 93.2 | 92.9 | 90.5 | 87.1 | 77 | 83.2 | 87.1 | 72.2 | 77.9 | 65.5 | **67.3** | 78.6 | 72.3 |
| SPoT | 76.8K | 85.4 | 90.1 | 93.0 | 93.4 | 90.0 | 79.7 | 69.8 | 57.1 | 82.3 | 74.0 | 77.2 | 67.0 | 50.0 | 46.4 | 62.9 |
| ATTEMPT | 232K | 84.3 | **90.3** | 93.0 | 93.2 | 89.7 | 85.7 | 73.4 | 57.4 | 83.4 | 74.4 | 78.8 | 66.8 | 53.8 | 78.6 | 70.5 |
| MPT | 77.6K | 85.9 | **90.3** | 93.1 | 93.8 | 90.4 | 89.1 | 79.4 | 62.4 | 85.6 | 74.8 | 79.6 | 69.0 | **67.3** | 79.8 | 74.1 |
| **BMTPT** (Ours) | 77.6K | 86.2$_{0.06}$ | **90.3**$_{0.32}$ | **93.4**$_{0.31}$ | **94.4**$_{0.04}$ | **90.9**$_{0.37}$ | 87.2$_{0.7}$ | **81.3**$_{1.48}$ | **86.6**$_{0.69}$ | 88.7 | 72.4$_{0.13}$ | 80.3$_{0.5}$ | 67.4$_{0.43}$ | **67.3**$_{0.00}$ | 85.7$_{1.87}$ | 74.6 |
| Fine-tuning* | \|LM\| | 85.7 | 91.1 | 92.0 | 92.5 | 88.8 | 90.2 | 75.4 | 54.9 | 83.8 | - | - | - | - | - | - |
| Adapters* | 1.9M | **86.3** | 90.5 | 93.2 | 93.0 | 89.9 | **90.2** | 70.3 | 61.5 | 84.4 | - | - | - | - | - | - |
| HyperFormer* | 280K | 85.7 | 90.0 | 93.0 | 94.0 | 89.7 | 87.2 | 75.4 | 63.7 | 84.8 | - | - | - | - | - | - |
| HyperDecoder* | 76.8K | 86.0 | **90.5** | **93.4** | 94.0 | 90.5 | 87.7 | 71.7 | 55.9 | 83.7 | - | - | - | - | - | - |
| ATTEMPT* | 232K | 83.8 | 90.0 | 93.1 | 93.7 | 90.8 | 86.1 | 79.9 | 64.3 | 85.2 | 74.4 | 78.3 | 66.5 | **69.2** | 82.1 | 74.1 |
| MPT* | 77.6K | 84.3 | 90.0 | 93.0 | 93.3 | 90.4 | 89.2 | **82.7** | 63.5 | 85.8 | **74.8** | 79.2 | **70.2** | 67.3 | **89.3** | **76.1** |
| **BMTPT*** (Ours) | 77.6K | 85.9$_{0.06}$ | 90.2$_{0.17}$ | 93.2$_{0.31}$ | **95.3**$_{0.04}$ | **91.2**$_{0.27}$ | 86.9$_{0.54}$ | 80.9$_{1.48}$ | 85.6$_{0.05}$ | 88.7 | 72.3$_{0.39}$ | 80.1$_{0.32}$ | 67.7$_{0.47}$ | 67.3$_{0.00}$ | **89.3**$_{0.00}$ | 75.3 |

Table 1: Experiment results for GLUE and SuperGLUE using T5-base, along with the number of trained parameters. BMTPT results are averaged across three runs, with subscripts indicating the standard deviation. The evaluation metrics are Pearson correlation for STS-B, F1 for MultiRC, and accuracy for the other tasks. Top rows use single-task adaptation with no parameter sharing during the target task adaptation, while bottom rows employ multi-task adaptation. The best performance among parameter-efficient fine-tuning methods is bolded. BMTPT consistently outperforms most baselines in GLUE and is comparable in SuperGLUE, affirming its robustness across language tasks.

| k-shot | Method | GLUE | | | | | | | | | SuperGLUE | | | | | |
|---|---|---|---|---|---|---|---|---|---|---|---|---|---|---|---|---|
| | | MNLI | QQP | QNLI | SST-2 | STS-B | MRPC | RTE | CoLA | Avg. | Multirc | BoolQ | WiC | WSC | CB | Avg. |
| 4 | PT | 40.1 | 63.2 | 40.4 | 53.0 | 88.8 | 68.1 | 56.3 | 27.4 | 54.7 | 61.8 | 61.6 | 51.2 | 60.4 | 53.5 | 57.7 |
| | MPT | **59.4** | 82.0 | 86.2 | 56.5 | 89.1 | 68.1 | **62.6** | 34.8 | 67.3 | **62.6** | 62.6 | 52.9 | 67.3 | 73.6 | 63.6 |
| | **BMTPT** (Ours) | 43.0 | **82.4** | **89.2** | 60.3 | 90.0 | 76.7 | 55.8 | 67.8 | 70.7 | 60.6 | **62.7** | 56.1 | 67.3 | 78.6 | 65.1 |
| 16 | PT | 41.5 | 62.3 | 59.9 | 50.9 | 87.8 | 68.1 | 54.7 | 28.5 | 56.7 | 60.3 | 61.9 | 48.9 | 44.2 | 63.5 | 55.8 |
| | MPT | 61.6 | 84.7 | 90.6 | 63.2 | 89.1 | 70.1 | **64.8** | 32.1 | 69.5 | **64.5** | 63.3 | 49.8 | 67.3 | 78.6 | 64.7 |
| | **BMTPT** (Ours) | 65.2 | 85.5 | 91.3 | 70.9 | 89.7 | 77.0 | 63.5 | 68.4 | 76.4 | 60.4 | 63.7 | 62.4 | 67.3 | 75.0 | 65.8 |
| 32 | PT | 37.0 | 62.3 | 56.7 | 50.9 | 87.5 | 68.1 | 54.7 | 23.2 | 55.1 | 59.2 | 61.7 | 52.6 | 67.3 | 67.8 | 61.7 |
| | MPT | 63.6 | 88.5 | 91.0 | 75.9 | 89.7 | 74.5 | **59.7** | 30.8 | 71.7 | **63.3** | 68.9 | 53.9 | 67.3 | 82.1 | 67.1 |
| | **BMTPT** (Ours) | 66.3 | 88.9 | 91.6 | 89.1 | 90.4 | 78.2 | 59.4 | 67.4 | 79.0 | 63.2 | 64.2 | 55.5 | 67.3 | 82.1 | 66.5 |

Table 2: Few-shot experiment results for GLUE and SuperGLUE using T5-base, using 4, 16, and 32 training instances. BMTPT results are averaged across three runs. In tasks with limited training data, BMTPT consistently surpasses MPT and prompt tuning.

| **BMTPT** Variations | GLUE | | | | | | | | | SuperGLUE | | | | | |
|---|---|---|---|---|---|---|---|---|---|---|---|---|---|---|---|
| | MNLI | QQP | QNLI | SST-2 | STS-B | MRPC | RTE | CoLA | Avg. | Multirc | BoolQ | WiC | WSC | CB | Avg. |
| Standard BMTPT | 86.2 | 90.3 | 93.4 | 94.4 | 90.9 | 87.2 | 81.3 | 86.6 | 88.7 | 72.4 | 80.3 | 67.4 | 67.3 | 85.7 | 74.6 |
| BMTPT w/ T5-large | 89.1 | 90.9 | 94.1 | 95.5 | 92.3 | 89.3 | 85.6 | 87.7 | 90.6 | 76.6 | 84.4 | 72.4 | 67.3 | 85.7 | 76.8 |
| BMTPT w/ T5-3B | 92.3 | 91.4 | 94.3 | 95.2 | 93.3 | 89.4 | 85.7 | 89.4 | 91.4 | 79.4 | 88.3 | 73.7 | 67.3 | 89.3 | 79.6 |
| Source task sampling (**1**) | 86.2 | 90.1 | 92.9 | 94.3 | 91.2 | 88.2 | 81.3 | 83.6 | 88.5 | 71.8 | 80.9 | 66.8 | 67.3 | 85.7 | 74.5 |
| Source task sampling (**2**) | 85.8 | 90.3 | 93.1 | 94.8 | 90.9 | 88.3 | 80.4 | 86.9 | 88.8 | 72.0 | 80.8 | 69.1 | 67.3 | 85.7 | 75.0 |
| w/ 10 particles | 85.7 | 90.4 | 93.3 | 93.8 | 90.8 | 90.8 | 77.0 | 85.2 | 88.4 | 72.7 | 78.9 | 68.6 | 67.3 | 83.1 | 74.1 |
| w/o prior | 85.3 | 87.1 | 93.0 | 94.3 | 90.9 | 88.4 | 79.7 | 83.9 | 87.8 | 71.9 | 78.2 | 66.8 | 67.3 | 82.1 | 73.3 |

Table 3: Table corresponding to Section 6.2. We examined BMTPT in larger models and evaluated three components of BMTPT: source task sampling, performance based on the number of particles, and the prior.

employ 5 particles for SVGD and use 6 source tasks as mentioned in Subsection 5.1, therefore forming a batch of size 30 ($5 \times 6$). Also we use $\sigma = 10^5$ for target adaptation loss, denoted at Eq. (6). For two-speed learning rate, we set 0.3 as the full-rank matrix learning rate and 0.4 as low-rank matrix learning rate. We use a batch of size 32 during target adaptation. For multi-target task adaptation, we first form a batch of input texts from target tasks, using example-proportional mixing strategy (Raffel et al., 2020b), then prepend a corresponding target prompt to each input text in the batch. We ran all the experiments three times using different random seeds and provide the mean and standard deviations of the results. In cases where a dataset lacks a publicly available test split with annotations, we adopt either the original development set as our test set or perform a split within the original development set to create separate development and test sets, following

Mahabadi et al. (2021).

**Baselines** We conduct a comprehensive comparison of BMTPT with various baseline methods, including full finetuning (FT), vanilla prompt tuning (PT) (Lester et al., 2021), existing prompt transfer methods such as SPoT (Vu et al., 2022), ATTEMPT (Asai et al., 2022) and MPT (Wang et al., 2023), as well as popular parameter-efficient approaches like Adapters (Houlsby et al., 2019) and BitFit (Ben Zaken et al., 2022). On GLUE, we additionally compare with several state-of-the-art multi-task learning methods including HyperFormer (Karimi Mahabadi et al., 2021b) and HyperDecoder (Ivison and Peters, 2022), along with multi-task variants of FT and Adapters. Also, to compare our algorithm with conventional multi-task transfer learning, we implement and evaluate a vanilla multi-task transfer method that learns a single prompt upon the combined loss of source tasks and transfers it to the target task. We either directly quote reported numbers or utilize publicly available source code under the same backbone for a fair comparison, as outlined in the respective papers (Mahabadi et al., 2021; Karimi Mahabadi et al., 2021b; Asai et al., 2022; Wang et al., 2023).

## 6 Results

In Section 6.1, we provide the main findings on GLUE and SuperGLUE benchmarks. For findings on MRQA and "Others" benchmarks, please refer to Appendix D. In Section 6.2, we further provide a set of analyses.

### 6.1 Main Results

#### 6.1.1 GLUE and SuperGLUE

As shown in the top part of Table 1, BMTPT achieves new state-of-the-art results in parameter-efficient fine-tuning for both GLUE and Super-GLUE, outperforming other prompt tuning transfer methods (Vu et al., 2022; Asai et al., 2022; Wang et al., 2023). Compared to vanilla PT (Lester et al., 2021), BMTPT demonstrates a relative improvement of 16.5% on GLUE and 16.8% on Super-GLUE. This highlights the advantages of transferring knowledge using Bayesian approach. It is worth mentioning that BMTPT outperforms the full fine-tuning baseline on both benchmarks, despite only tuning 0.035% of the parameters compared to full fine-tuning.

The results presented in the bottom part of Table 1 demonstrate the ability of BMTPT to effectively utilize multi-task knowledge during fine-tuning on a group of target tasks. This highlights that BMTPT can benefit from multi-target adaptation setting, by further reducing the number of trainable parameters.

We also compare the performance of BMTPT and vanilla multi-task transfer that is introduced in Section 5.2, in Table 1. Surprisingly, vanilla multi-task transfer shows strong performance in GLUE and SuperGLUE tasks, outperforming competitive baselines. This result supports Section 2.1 which claims that previous methods (Vu et al., 2022; Asai et al., 2022; Wang et al., 2023) are not the optimal transfer technique. It is worth noting that BMTPT outperforms vanilla multi-task transfer. To understand this advantage, we may delve into the Bayesian perspective of BMTPT, which includes conventional transfer learning. While vanilla multi-task transfer only learns an initialization point that contains relatively limited source task information (Shwartz-Ziv et al., 2022), BMTPT learns posterior from the source tasks and adopts it as prior during target adaptation, enabling a richer and more insightful adaptation process.

#### 6.1.2 Few-Shot Experiments

We also present the results of the few-shot experiments on the GLUE and SuperGLUE datasets. For the 4-shot experiments, the learning rates were reduced to one-third of their original values to accommodate the decreased batch size relative to standard experiments. The performance figures for BMTPT are averaged over three runs, each initialized with a different random seed. These outcomes suggest that the prior used in target adaptation effectively positions the prompts to an optimal initial point for task adaptation in low-resource conditions.

### 6.2 Analyses

#### 6.2.1 Model Scaling

We perform scaling experiments to analyze the performance of BMTPT as the size of the pre-trained model increases. The result demonstrates that BMTPT can largely benefit from scaling LM to larger models. This aligns with the finding by (Lester et al., 2021), which suggests that prompt tuning is effective especially when applied to larger backbone LMs. Note that BMTPT achieves comparable performance to fully fine-tuned models even

with T5-base, meaning that BMTPT is effective across various model scales.

### 6.2.2 Effectiveness of Source Task Sampling

To evaluate the effectiveness of Source Task Sampling discussed in Section 4.2, we conducted experiments under two settings: (**1**) subsampling 3 tasks from a pool of 6 source tasks (refer to Section 5.1), to examine if Source Task Sampling can mitigate performance degradation at limited computation resource scenario, and (**2**) diversifying the source task set to include 12 tasks and subsampling 6 tasks from this expanded set, to investigate the potential benefits of Source Task Sampling with an expanded source task set. For the second setting, we expand the source task set with AGNews (Zhang et al., 2015), CommonsenseQA (Talmor et al., 2019), OpenBookQA (Mihaylov et al., 2018), ARC (Clark et al., 2018), adversarial NLI (Nie et al., 2020), and Winogrande (Sakaguchi et al., 2020).

From Table 3, we can see that setting (**1**) shows minimal performance degradation compared to the case with 6 source tasks. This finding indicates the successful application of the Source Task Sampling technique in low computation resource scenarios. Also, setting (**2**) demonstrates slight performance enhancements, suggesting that Source Task Sampling can derive benefits from diversifying the source task set.

### 6.2.3 BMTPT Performance on Different Numbers of Particles

Since SVGD is a particle-based VI method, the number of particles employed may affect the performance of our method. Therefore we investigate the effect of the number of particles on target adaptation performance by comparing 5-particle BMTPT and 10-particle BMTPT (Table 3). We found that the 10-particle case does not yield better results than the 5-particle case. Because of the instability reported in the original SVGD paper (Liu and Wang, 2016) and a similar empirical finding from Yoon et al. (2018) we speculate that this absence of enhancement might be attributed to the inherent characteristics of SVGD, including its sensitivity to kernel function parameters.

### 6.2.4 Effect of Prior

To assess the impact of the prior term in Eq. (6), we conducted an ablation experiment by removing the prior term from the target adaptation loss. The ablated version of BMTPT exhibited poorer performance, implying the efficacy of learning an informative source posterior and leveraging it during target adaptation to facilitate effective transfer learning.

## 7 Conclusion

We present Bayesian Multi-Task Prompt Tuning (BMTPT), a Bayesian approach for transferring soft prompt. Our method defines a posterior distribution over prompt on source tasks, and approximates the posterior using SVGD, then initializes the target prompt with aggregation of source prompts while regularizing the training of the target prompt using transferred posterior. Empirically we found this approach achieves comparable or superior performance over strong parameter-efficient fine-tuning baselines.

**Limitations** While showing compelling experimental results with only the use of a soft prompt, BMTPT has its limitations. Primarily, appending the soft prompt to the input text leads to an extension in the overall input length, consequently increasing the memory footprint. This is a well-known issue in PT (Karimi Mahabadi et al., 2021a), and BMTPT is not immune to it. Furthermore, in BMTPT, since multiple particles are used and source task sentences are appended to each, the batch size grows to be a multiple of the number of particles. This could potentially increase memory demands during the source posterior learning. However, this can be mitigated by implementing Source Task Sampling or reducing the number of particles. Experiments determining the optimal number of particles have not been performed in our study, and future research could potentially explore this aspect to ascertain the most appropriate number of particles.

**Acknowledgements** This work was supported by the "Development of Efficient Fine-Tuning and Zero-Shot Generalization Methods" project funded by KT (KT award B220002586), IITP grant funded by MSIT (No.2019-0-00075, AI Graduate School Program (KAIST); No.2020-0-00940, Foundations of Safe Reinforcement Learning and Its Applications to Natural Language Processing; No.2022-0-00311, Development of Goal-Oriented Reinforcement Learning Techniques for Contact-Rich Robotic Manipulation of Everyday Objects), Artificial intelligence

industrial convergence cluster development project funded by the Ministry of Science and ICT(MSIT, Korea) & Gwangju Metropolitan City, NRF of Korea (NRF2019R1A2C1087634), Field-oriented Technology Development Project for Customs Administration through NRF of Korea funded by the MSIT and Korea Customs Service (NRF2021M3I1A1097938), ETRI grant (22ZS1100, Core Technology Research for Self-Improving Integrated AI System), KAIST-NAVER Hypercreative AI Center.

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

## A  Analogy with Gaussians for the Aggregation of Prompts Trained on Diverse Tasks

Consider a scenario with $K$ source tasks, each characterized by a posterior $p(\boldsymbol{\theta}|\mathcal{D}_k) = \mathcal{N}(\boldsymbol{\mu}_k, \boldsymbol{\Lambda}_k^{-1})$, where $\mathcal{D}_k$ represents the dataset of $k$-th task, and $\boldsymbol{\theta}$ is a soft prompt. Under the uniform prior, maximizing the likelihood (MLE) is equivalent to MAP estimation, and would lead each source prompt trained on task $k$ to the mode $\boldsymbol{\mu}_k$. By combining individual posteriors with assuming independent selection of tasks, we can construct the global posterior $p(\boldsymbol{\theta}|\mathcal{D} \equiv \bigcup_{k=1}^K \mathcal{D}_k) \propto \prod_{k=1}^K p(\boldsymbol{\theta}|\mathcal{D}_k)$[3]. The goal of transfer learning is to maximize this posterior, anticipating that the overall knowledge captured from source tasks will lead to a good starting point for a target task. Note that the posterior, which is a product of Gaussian distributions, is a Gaussian distribution whose mean is $\boldsymbol{\mu}_{\text{global}} = (\sum_{k=1}^K \boldsymbol{\Lambda}_k)^{-1} (\sum_{k=1}^K \boldsymbol{\Lambda}_k \boldsymbol{\mu}_k)$. Since the mean is the mode of a Gaussian, $\boldsymbol{\mu}_{\text{global}}$ would be a good candidate for the initialization point of the target prompt. However, unless the covariances differ only by a scaling factor, a weighted sum of the individual modes $\{\boldsymbol{\mu}_k\}_{k=1}^K$ is unlikely to equal $\boldsymbol{\mu}_{\text{global}}$.

## B  Details for SVGD

### B.1  Choice of SVGD

Stein Variational Gradient Descent (SVGD) is a nonparametric variational inference technique that amalgamates the benefits of Markov Chain Monte Carlo (MCMC) and variational inference (Liu and Wang, 2016). Our utilization of SVGD over conventional variational inference (VI) methods is driven by multiple factors, each rooted in the limitations and attributes of standard VI approaches.

The target posterior distribution we aim to approximate is complex, potentially even multi-modal. Standard VI methods, constrained by a specific family of distributions, often fail to capture such intricate structures. Therefore, they can show an inherent bias toward particular tasks. In contrast, SVGD employs a particle-based approach to dynamically generate a more expansive class of approximating distributions. This capability allows SVGD to represent complex and multi-modal distributions with greater accuracy.

Furthermore, traditional VI methods like Variational Autoencoders (VAE) are generator-based and necessitate sampling. In contrast, SVGD requires the log derivatives of the prior at each point, commonly referred to as the score function. Additionally, while most VI methods aim to minimize surrogates of KL divergence through optimization, SVGD employs a first-order update method with a competing mechanism between particle repulsion and gradient descent.

### B.2  Mathematical Explanation

Whereas gradient descent guides particles towards the optimal direction of fastest objective decrease, SVGD identifies the optimal transformation to minimize the KL divergence between the current and target distributions.

To find the optimal direction in the unit ball $\mathcal{B}$ of the Reproducing Kernel Hilbert Space $\mathcal{H}$, which is the closed linear span of $\{k(\boldsymbol{\theta}, \cdot) : \boldsymbol{\theta} \in \mathbb{R}^D\}$, that minimizes the KL-divergence towards the target distribution $p$, SVGD uses the point transformation $\mathbb{T}_{[\alpha\boldsymbol{\phi}]}(\boldsymbol{\theta}) = (\mathbf{I} + \alpha\boldsymbol{\phi})(\boldsymbol{\theta})$. We will use the same notation for probability density with probability measure $\mu$, if there is no confusion. Specifically, it finds $\boldsymbol{\phi}^*$ that satisfies:

$$\frac{\boldsymbol{\phi}^*_{\mu,p}}{\|\boldsymbol{\phi}^*_{\mu,p}\|_{\mathcal{H}}} = \underset{\boldsymbol{\phi} \in \mathcal{B}}{\arg\max} \left\{ -\frac{d}{d\alpha} \text{KL}\big(\mathbb{T}_{[\alpha\boldsymbol{\phi}]} \# \mu \,\|\, p\big)\Big|_{\alpha=0} \right\},$$

where $\mathbb{T}\#\mu(A) = \mu(\mathbb{T}^{-1}(A))$[4]. The closed-form solution of the above is given by:

$$\boldsymbol{\phi}^*_{\mu,p} = \int_{\mathbb{R}^d} \big[\nabla \log p(\boldsymbol{\theta}) k(\boldsymbol{\theta}, \cdot) + \nabla k(\boldsymbol{\theta}, \cdot)\big] \mu(d\boldsymbol{\theta}).$$

Here, $\log p(\boldsymbol{\theta})$ is the log-likelihood of $p$. The SVGD algorithm updates the distribution as follows:

$$\mu_{t+1} = (I + \alpha\boldsymbol{\phi}^*_{\mu,p})\#\mu_t,$$

---

[3]Under the uniform prior assumption, this relation can be derived from $p(\boldsymbol{\theta}|\mathcal{D}_k) \propto p(\mathcal{D}_k|\boldsymbol{\theta})$ for each $k$. Please refer the Remark in the Section 3.

[4]If random variable $X$ follow distribution $\mu$, the distribution $\mathbb{T}\#\mu$ can be seen as the distribution of $\mathbb{T}(X)$.

where $\alpha$ is the step size. Discretized version of the above update rule for a finite set of particles $\{\boldsymbol{\theta}_i\}_{i=1}^M$, SVGD iteratively transports the particles using the following update rule for $m \in [M]$:

$$\boldsymbol{\theta}_i \leftarrow \boldsymbol{\theta}_i + \alpha\boldsymbol{\phi}^*(\boldsymbol{\theta}_i), \text{where } \boldsymbol{\phi}^*(\boldsymbol{\theta}_i) \text{ is } \frac{1}{M}\sum_{j=1}^M [\nabla_{\boldsymbol{\theta}_j} \log p(\boldsymbol{\theta}_j)k(\boldsymbol{\theta}_j, \boldsymbol{\theta}_i) + \nabla_{\boldsymbol{\theta}_j} k(\boldsymbol{\theta}_j, \boldsymbol{\theta}_i)].$$

The behavior inherent to SVGD is orchestrated by the two terms in the update, which define the key control mechanisms. Firstly, the first term entails the sharing of gradient information among particles, guiding their update trajectory. Additionally, the influence of neighboring particles is modulated by kernel distance weighting. The second term, $\nabla_{\boldsymbol{\theta}_j} k(\boldsymbol{\theta}_j, \boldsymbol{\theta}_i)$, introduces a repelling force between the particles, preventing them from converging to a single mode.

### B.3  Detailed Explanation for RBF Kernel

In the execution of the Stein Variational Gradient Descent (SVGD) for our set of particles denoted as $\{\boldsymbol{\theta}_i\}_{i=1}^M$, we adopted the Radial Basis Function (RBF) kernel, which is defined as follows:

$$k(\boldsymbol{\theta}_1, \boldsymbol{\theta}_2) = \exp\left(-\frac{\|\boldsymbol{\theta}_2 - \boldsymbol{\theta}_1\|^2}{h}\right), \text{where } h = \frac{\left(\text{Median}\left\{\|\boldsymbol{\theta}_j - \boldsymbol{\theta}_i\|^2 \,\middle|\, i \neq j, \, i, j \in [M]\right\}\right)^2}{\log(M+1)}.$$

In this formulation, $h$ is a parameter frequently adjusted according to the distances between particles. As part of our methodology, we adhere to the median heuristic, a strategy supported by previous studies (Schölkopf and Smola, 2018; Ba et al., 2022). This entails designating the bandwidth as the median of the set of mutual distances between particles.

### B.4  Damped SVGD

The variant of Stein Variational Gradient Descent (SVGD) we employ in this work is Damped SVGD, as delineated in the work by Ba et al. (2022). SVGD, in its typical implementation, is prone to variance collapse when applied in a finite regime with particles, rather than updating distributions directly. This necessitates an adaptation of the SVGD's update rule to ensure a proper approximation of the distribution with particles. The Damped SVGD specifically addresses this issue by moderating the influence of its own gradient descent term. This adjustment can be seen clearly in the update rule for $\boldsymbol{\theta}_i$ in a configuration $\{\boldsymbol{\theta}_i\}_{i=1}^M$. As compared to the standard update rule, the modification reads:

$$\boldsymbol{\phi}^*_{\text{damped}}(\boldsymbol{\theta}_i) = \frac{1}{M}\sum_{j \neq i} [\nabla_{\boldsymbol{\theta}_j} \log p(\boldsymbol{\theta}_j)k(\boldsymbol{\theta}_j, \boldsymbol{\theta}_i) + \nabla_{\boldsymbol{\theta}_j} k(\boldsymbol{\theta}_j, \boldsymbol{\theta}_i)] + \frac{1}{M}\lambda \cdot \nabla_{\boldsymbol{\theta}_i} \log p(\boldsymbol{\theta}_i)k(\boldsymbol{\theta}_i, \boldsymbol{\theta}_i)$$

$$= \boldsymbol{\phi}^*(\boldsymbol{\theta}) - (1 - \lambda)\frac{1}{M}\nabla_{\boldsymbol{\theta}_i} \log p(\boldsymbol{\theta}_i)k(\boldsymbol{\theta}_i, \boldsymbol{\theta}_i)$$

$$= \boldsymbol{\phi}^*(\boldsymbol{\theta}) - \frac{1 - \lambda}{M}\nabla_{\boldsymbol{\theta}_i} \log p(\boldsymbol{\theta}_i).$$

In the Damped SVGD paper, the parameter $\lambda$ can be chosen using one of two strategies: taking $\lambda$ as $\lambda_{\min} = \min\left(1, e^{-1}(1 + \frac{M}{l \cdot d})\right)$ for "fully damped," or taking $\lambda$ as a value between $\lambda_{\min}$ and 1 for "intermediate." In our experiments, we use "intermediate" by consistently choosing the value $\min\left(1, e^{-1}(5 + \frac{M}{l \cdot d})\right)$, taking both selections into account. In our standard setting, this yields $\lambda \approx 0.368$. This variant of SVGD improves upon the original by mitigating the issue of variance collapse in some degree.

## C  Computational Complexity Analysis of BMTPT

Our computational analysis verifies that BMTPT is computationally efficient for both source task posterior learning and task adaptation stages. Note that the additional computation necessary for BMTPT occurs after the prompt receives the back-propagated gradient information from the LLM.

## C.1 Definitions of Notations

- $M$: Number of particles.

- $l$: Length of the prompt.

- $d$: Hidden dimension of LLM (Large Language Model).

- $d_{\text{prompt}} = d \times l$: Dimension of the prompt.

- $T_{\text{grad}}$: Number of operations for the gradient backpropagation through the backbone LLM.

- $\mathbf{K}$: RBF Kernel matrix with dimensions $M \times M$ ($\mathbf{K}_{i,j} = k(\boldsymbol{\theta}_i, \boldsymbol{\theta}_j)$).

- $\boldsymbol{\Theta}$: Matrix of prompt parameters with dimensions $M \times d_{\text{prompt}}$.

- $\nabla \log \mathbf{p}$: Gradient of log-probability for each particle.

- $\alpha$: Stepsize.

## C.2 Source Task Training

In the source task training phase, we have a multi-particle formulation governed by SVGD with an RBF Kernel. The formulation involves various matrix and vector products, which we denote as

$$\Delta\boldsymbol{\Theta} = \alpha\Big(\mathbf{K}\nabla\log\mathbf{p} + \frac{2}{h}(\text{diag}(\mathbf{K1}) - \mathbf{K})\boldsymbol{\Theta}\Big).$$

The computational complexity for BMTPT during this phase can be summarized as $\mathcal{O}(T_{\text{grad}}) + M^2 \cdot \mathcal{O}(d_{\text{prompt}})$. This indicates that BMTPT requires additional $M^2 \cdot \mathcal{O}(d_{\text{prompt}})$ calculations over the vanilla prompt tuning. However, since $T_{\text{grad}}$ is the dominating factor and $M^2 = 25$ in our experiments, this increase is computationally acceptable. The average wall-clock time recorded during the training of the source task, based on 5 updates, is as follows: 0.42 seconds for the backward pass through the language model (LM) and 0.0035 seconds for Damped SVGD. We used a single GeForce RTX 3090 GPU for these computations.

## C.3 Task Adaptation Stage

During the task adaptation stage, the additional computational complexity of BMTPT is mainly due to the upper bound of $\log$ prior term, which takes $\mathcal{O}(d_{\text{prompt}})$ computations. Therefore, the computational complexity for a single update is $\mathcal{O}(T_{\text{grad}}) + \mathcal{O}(d_{\text{prompt}})$. Here as well, the dominating factor is $T_{\text{grad}}$. Similarly, we report the wall-clock time observed on our device during the target adaptation phase, specifically for the SuperGLUE-CB task with a batch size of 32. The forward pass through the LM takes an average of 0.16 seconds, while the forward pass for the prior term requires 0.00011 seconds. We used a single GeForce RTX 3090 GPU.

## D Experiment on MRQA and "Others" Benchmark

| Method | # Params | MRQA | | | | | Others | | | | |
|---|---|---|---|---|---|---|---|---|---|---|---|
| | | NQ | HP | SQA | News | Avg. | WG | Yelp | SciTail | PAWS | Avg. |
| Fine-tuning | \|LM\| | 75.1 | 77.5 | 81.1 | 65.2 | 74.7 | 61.9 | 96.7 | 95.8 | 94.1 | 87.1 |
| Adapter | 1.9M | 74.2 | 77.6 | 81.4 | 65.6 | 74.7 | 59.2 | 96.9 | 94.5 | 94.3 | 86.2 |
| BitFit | 280K | 70.7 | 75.5 | 77.7 | 64.1 | 72.0 | 57.2 | 94.7 | 94.7 | 92.0 | 84.7 |
| PT | 76.8K | 67.9 | 72.9 | 75.7 | 61.1 | 69.4 | 49.6 | 95.1 | 87.9 | 55.8 | 72.1 |
| SPoT | 76.8K | 68.2 | 74.8 | 75.3 | 58.2 | 69.1 | 50.4 | 95.4 | 91.2 | 91.1 | 82.0 |
| ATTEMPT | 232K | 70.4 | 75.2 | 77.3 | 62.8 | 71.4 | 57.6 | 96.7 | 93.1 | 92.1 | 84.9 |
| MPT | 77.6K | 72.0 | 75.8 | 77.2 | 63.7 | 72.2 | 56.5 | 96.4 | 95.5 | 93.5 | 85.5 |
| **BMTPT** (Ours) | **77.6K** | $69.6_{0.21}$ | $82.9_{0.22}$ | $76.2_{0.09}$ | $62.4_{0.07}$ | 72.8 | $55.6_{0.37}$ | $97.6_{0.03}$ | $95.4_{0.47}$ | $93.7_{0.22}$ | 85.5 |

Table 4: Experiment results on MRQA and Others. We evaluate MRQA tasks using F1 score and Others using accuracy. BMTPT results are averaged over three runs with standard deviation indicated by subscripts.

For MRQA and several "Others" datasets, BMTPT remains competent among parameter-efficient baselines, showing the versatility of our approach outside GLUE and SuperGLUE.