# OpenReview forum: "Bayesian Multi-Task Transfer Learning for Soft Prompt Tuning"
_EMNLP/2023/Conference — EMNLP 2023 Findings_

### Official Review · Reviewer_f8vf · 2023-08-05

**Soundness:** 3

**Excitement:**

3: Ambivalent: It has merits (e.g., it reports state-of-the-art results, the idea is nice), but there are key weaknesses (e.g., it describes incremental work), and it can significantly benefit from another round of revision. However, I won't object to accepting it if my co-reviewers champion it.

**Paper Topic And Main Contributions:**

As the paper’s name suggests, this paper proposes a Bayesian approach to conduct transfer learning from multiple separately trained soft prompts from other source tasks to target tasks as the target soft prompt initialization. In this method, representative source prompts are calculated corresponding to the samples from the posterior utilizing Stein Variational Gradient Descent, which are then aggregated to constitute the initial target prompt.

**Reasons To Accept:**

1. Clear writing of the paper.
2. Thorough experimental analysis that tried multiple model sizes and various source/target tasks.

**Reasons To Reject:**

1. My main concern of the paper is that the proposed method doesn’t show a significant improvement compared to other existing methods in Table 1 lower part.
2. I don’t quite understand the motivation of the method. Intuitively, what are the expected benefits of the proposed method compared to existing methods?

**Reproducibility:**

3: Could reproduce the results with some difficulty. The settings of parameters are underspecified or subjectively determined; the training/evaluation data are not widely available.

**Reviewer Confidence:**

4: Quite sure. I tried to check the important points carefully. It's unlikely, though conceivable, that I missed something that should affect my ratings.

---

> ### Author Rebuttal · Authors · 2023-08-29
>
> ### **Clarification on Soft Prompts and Multi-Source Data**
> Thank you for your comprehensive review and for acknowledging the clarity and thoroughness of our paper. We appreciate your recognition of our Bayesian approach, which utilizes SVGD for representative source prompts. However, we would like to point out that each soft prompt learns the joint posterior distribution from multi-source data, rather than assigning each source task to a specific prompt. These prompts are then aggregated to serve as the initial target prompt, offering a systematic method for transfer learning from multiple source tasks to target tasks.
>
> ### **Performance Relative to Baselines**
> We acknowledge the reviewer's observation regarding the relative performance of our proposed BMTPT, as highlighted in Table 1. It is imperative to note that while the model may not have shown a pronounced advantage over the baselines in certain tasks, its robust performance across multiple benchmarks is noteworthy. Specifically, the BMTPT either outperforms or matches existing baselines on a significant number of tasks within the GLUE/SuperGLUE framework. This generalized performance reflects the model's reliability across various linguistic tasks, serving as an important metric beyond mere task-specific performance. Please also refer to the **Empirical Performance** section in the initial response to reviewer *qA2o*.
>
> We wish to point out that our method performs well in few-shot scenarios, as shown in the initial response to reviewer *g8bk*. The result indicates both effective initial task adaptation and increased efficiency in real-world conditions.
>
> In the previous version of the manuscript, we inadvertently presented the performance metrics for multi-task ATTEMPT rather than MPT in Table 2. Below, we provide the corrected Table 2 with the performance metrics for MRQA and "Others" datasets for the MPT model:
>
> ### Result on MRQA Datasets
> | Method        | params | NQ    | HP    | SQA   | News  | Avg.  |
> |---------------:|:-------:|:-------:|:-------:|:-------:|:-------:|:-------:|
> | Adapter       | 1.9M | **74.2**  | 77.6  | **81.4**  | **65.6**  | **74.7**  |
> | BitFit        | 280K | 70.7  | 75.5  | 77.7  | 64.1  | 72.0  |
> | Prompt tuning | 76.8K | 67.9  | 72.9  | 75.7  | 61.1  | 69.4  |
> | SPoT          | 76.8K | 68.2  | 74.8  | 75.3  | 58.2  | 69.1  |
> | ATTEMPT       | 232K | 70.4  | 75.2  | 77.3  | 62.8  | 71.4  |
> | MPT           | 77.6K | 72.0  | 75.8  | 77.2  | 63.7  | 72.2  |
> | BMTPT (Ours)  | 77.6K | 69.6  | **82.9**  | 76.2  | 62.4  | 72.8  |
>
> ### Result on  ''Others'' Datasets
> | Method        | params | WG    | Yelp  | SciTail | PAWS  | Avg.  |
> |---------------:|:-------:|:-------:|:-------:|:-------:|:-------:|:-------:|
> | Adapter       | 1.9M | **59.2**  | 96.9  | 94.5    | **94.3**  | **86.2**  |
> | BitFit        | 280K | 57.2  | 94.7  | 94.7    | 92.0  | 84.7  |
> | Prompt tuning | 76.8K | 49.6  | 95.1  | 87.9    | 55.8  | 72.1  |
> | SPoT          | 76.8K | 50.4  | 95.4  | 91.2    | 91.1  | 82.0  |
> | ATTEMPT       | 232K | 57.6  | 96.7  | 93.1    | 92.1  | 84.9  |
> | MPT           | 77.6K | 56.5  | 96.4  | **95.5**    | 93.5  | 85.5  |
> | BMTPT (Ours)  | 77.6K | 55.6  | **97.6**  | 95.4 | 93.7  | 85.6  |
>
> Note that BMTPT significantly improves the average performance of PT by +3.4% on MRQA and +13.5% on the "Others" benchmark.
>
> ### **Fundamental Motivation Behind BMTPT**
> We appreciate the query about the fundamental motivation behind our method. **At its core, BMTPT aims to focus on the essence of transfer learning by enhancing it via transferring a useful distribution as a prior for target adaptation. On the other hand, existing prompt transfer methods such as SPoT, ATTEMPT, and MPT tend to rely on the transferability between individual NLP tasks (e.g., SQuAD is more helpful for solving MRPC than SST-2)**. By employing a Bayesian framework, we leverage SVGD updates and prior injecting objectives, which are computationally efficient and straightforward to implement.
>
> ### **Contribution and Applicability**
> This Bayesian approach allows for the optimization for various NLP tasks without requiring additional neural network structures or procedures like distillation (ATTEMPT uses an attention module and MPT utilizes a separate backbone model for distillation). Consequently, our model provides a unified and more efficient method that performs competitively or surpasses existing approaches including other PEFT methods and full fine-tuning. We believe this represents a significant contribution to the field, substantiating both the theoretical significance and practical applicability of our method.

---

### Official Review · Reviewer_g8bk · 2023-08-12

**Typos Grammar Style And Presentation Improvements:** 1. Boldening the highest numbers in t…
**Soundness:** 4

**Excitement:**

4: Strong: This paper deepens the understanding of some phenomenon or lowers the barriers to an existing research direction.

**Paper Topic And Main Contributions:**

The paper introduces BMTPT as a a novel Bayesian approach to multi-task prompt tuning. In contrast to previous approaches in the literature that learn source prompts individually, BMTPT learns a global distribution over the source prompts which enables better capturing the correlations between these tasks. The approach then initializes and learns the target task prompts from the source prompts. Results on transferring from 6 source tasks to 21 target tasks show that the method outperform existing parameter efficient fine tuning methods in the literature.

**Questions For The Authors:**

1. I am curious if the authors could include precise params/task numbers like ATTEMPT and MPT do in their papers. While section 4.4 computes this parametrically, having the exact values (especially in the table) would enhance comparisons.

2. I am wondering about the choice of SGVD over regular variational inference, given its complexity. I am curious to know whether the authors explored using VI instead of SVGD. If they did, how did it perform? Additionally, I am curious to know how the optimization process might differ with VI. I assume that the only thing that would change is learning the posterior of source task prompts and drawing samples from $p(. | \mathcal{D}^\mathcal{S})$ in learning target prompts?

3. I was wondering if the authors have done experiments in the few-shot scenario?

4. I'm wondering if the authors could include comparisons with other methods when scaling up to larger models like T5-large and T5-3B.

**Reasons To Accept:**

1. Acceptable results showing that the method outperforms the existing PEFT methods on GLUE.
2. This work can encourage more work on Bayesian multi-task prompt training in the future.
3. The paper is clearn and well-written.

**Reasons To Reject:**

1. The multi-task results on Super-GLUE are worse than MPT.
2. The method does not compare to existing work in the literature when scaling the model in section 6.2.

**Reproducibility:**

4: Could mostly reproduce the results, but there may be some variation because of sample variance or minor variations in their interpretation of the protocol or method.

**Reviewer Confidence:**

3: Pretty sure, but there's a chance I missed something. Although I have a good feel for this area in general, I did not carefully check the paper's details, e.g., the math, experimental design, or novelty.

---

> ### Author Rebuttal · Authors · 2023-08-29
>
> We appreciate your comprehensive evaluation and the acknowledgment of our study's contributions. It's gratifying to see that BMTPT geared toward understanding a unified distribution for source prompts, was identified as valuable for its efficacy in correlating tasks.
>
> ### **Regarding the Number of Tunable Parameters**
> We emphasize that our method is parameter-efficient, using nearly the same number of parameters as MPT. We also highlight the overall GLUE performance and the number of parameters as follows:
>
> ### **Table for Average Sigle-Target Performance on GLUE Datasets**
> | Method        | FT | Adapters   | BitFit  | PT  | Vanila Transfer PT | SPoT  | ATTEMPT | MPT | BMTPT  |
> |---------------:|:-------:|:-------:|:-------:|:-------:|:-------:|:-------:|:-------:|:-------:|:-------:|
> | #params   | 220M | 1.9M  | 280K | 76.8K | 76.8K | 76.8K | 232K  | 77.6K | 77.6K |
> | GLUE Avg.  | 84.9 | 84.5  | 83.3  | 72.2  | 87.1  | 82.3 | 83.4 | 85.6 | **88.7**  |
>
> While our multi-task performance on SuperGLUE is marginally lower than that of MPT, the overall trend in empirical results is notable, particularly in the context of parameter efficiency.
>
> ### **Reason for using SVGD**
> We appreciate your inquiry regarding our decision to use Stein Variational Gradient Descent (SVGD) over traditional variational inference methods. Our choice is motivated by several considerations that emerge from the limitations and features of standard variational inference methodologies.
>
> Firstly, the source task posterior we aim to approximate possesses a complex, perhaps multi-modal distribution. Conventional variational inference, restricted by distribution families, often fails to capture such complexity adequately, and can be overly biased towards specific task. On the other hand, SVGD adaptively generates a richer class of approximating distributions using particles. This enables SVGD to approximate complex and multi-modal target distributions with higher fidelity.
>
> Furthermore, the theoretical foundations of SVGD offer an additional layer of reliability. SVGD framework is originated from the first-order optimization on the functional space, which provides solid backgrounds for the optimization strategy.
>
> In response to the reviewer's question, we clarify the distinctions between traditional VI and SVGD. Traditional VI methods like Variational Autoencoders (VAE) are generator-based and necessitate sampling. In contrast, SVGD requires the log derivatives of the prior at each point, commonly referred to as the score function. Additionally, while most VI methods aim to minimize surrogates of KL divergence through optimization, SVGD employs a first-order update method with a trade-off mechanism between particle repulsion and gradient descent.
>
> ### **Few-Shot Learning Experiments on GLUE and SuperGLUE**
> In this section, we present the requested few-shot experiments conducted on the GLUE and SuperGLUE datasets. For the 4-shot experiments, we adjusted the learning rates to one-third of their original values, to account for the reduced batch size compared to our standard experiments. The reported performance for BMTPT is averaged over three runs, each operated with a different random seed.
>
> These results demonstrate that our prior in target adaptation guides the prompts toward an efficient initial point for task adaptation in low-resource scenario.
>
> ### 4-shot (GLUE)
> Method | MNLI | QQP | QNLI | SST-2 | STS-B | MRPC | RTE | CoLA | Avg. |
> |--------|------|-----|------|-------|-------|------|-----|------|------|
> PT     | 40.1 | 63.2| 40.4 | 53.0  | 88.8  | 68.1 | 56.3| 27.4 | 54.7 |
> MPT    | **59.4** | 82.0| 86.2 | 56.5  | 89.1  | 68.1 | **62.6**| 34.8 | 67.3 |
> BMTPT(Ours)  | 43.0 | **82.4** | **89.2** | **60.3** | **90.0** | **76.7** | 55.8 | **67.8** | **70.65** |
>
> ### 16-shot (GLUE)
>  Method | MNLI | QQP | QNLI | SST-2 | STS-B | MRPC | RTE | CoLA | Avg. |
> |--------|------|-----|------|-------|-------|------|-----|------|------|
>  PT     |  41.5 | 62.3| 59.9 | 50.9  | 87.8  | 68.1 | 54.7| 28.5 | 56.7 |
> MPT    |61.6 | 84.7| 90.6 | 63.2  | 89.1  | 70.1 | **64.8**| 32.1 | 69.5 |
> BMTPT(Ours)  | **65.2** | **85.5** | **91.3** |	**70.9** | **89.7** | **77.0** | 63.5 |	**68.4** | **76.4** |
>
> ### 32-shot (GLUE)
>  Method | MNLI | QQP | QNLI | SST-2 | STS-B | MRPC | RTE | CoLA | Avg. |
> |--------|------|-----|------|-------|-------|------|-----|------|------|
>  PT     |37.0|62.3|56.7|50.9|87.5|68.1|54.7|23.2|55.1|
> MPT    |63.6|88.5|91.0|75.9|89.7|74.5|**59.7**|30.8|71.7|
> BMTPT(Ours)  |**66.3**|**88.9**|**91.6**|**89.1**|**90.4**|**78.2**|59.4|**67.4**|**79.0**|
>
> ### 4-shot (SuperGLUE)
> Method | Multi | BoolQ | WiC | WSC | CB | Avg. |
> |--------|------|-----|------|-------|------|------|
> PT     | 61.8 | 61.6 | 51.2 | 60.4  | 53.5  | 57.7 |
> MPT    | **62.6** | 62.6 | 52.9 | **67.3**  | 73.6  | 63.6 |
> BMTPT(Ours)  | 60.6|**62.7**|**56.1**|**67.3**|**78.6**|**65.1**|
>
> ### 16-shot (SuperGLUE)
> Method | Multi | BoolQ | WiC | WSC | CB | Avg. |
> |--------|------|-----|------|-------|------|------|
> PT     | 60.3 | 61.9 | 48.9 | 44.2 | 63.5 | 55.8 |
> MPT    | **64.5** | 63.3 | 49.8 | **67.3** | **78.6** | 64.7 |
> BMTPT(Ours)  |60.4	| **63.7**| **62.4**| **67.3** |75.0 |**65.8**|
>
> ### 32-shot (SuperGLUE)
> Method | Multi | BoolQ | WiC | WSC | CB | Avg. |
> |--------|------|-----|------|-------|------|------|
> PT     | 59.2 | 61.7 | 52.6 | **67.3** | 67.8 | 61.7 |
> MPT    | **63.3** | **68.9** | 53.9 | **67.3** | **82.1** | **67.1** |
> BMTPT(Ours)  |63.2|64.2|**55.5**|**67.3**|**82.1**|66.5|
>
> ### **Comparison on the Larger Models with Baselines**
> Our experimental results on GLUE and SuperGLUE, using T5-large and T5-3B, are detailed in Table 3. A direct performance comparison with ATTEMPT and MPT, cutting-edge prompt tuning methods, could not be facilitated due to the absence of specific performance metrics in their respective papers (see Figure 4 in both ATTEMPT and MPT). Instead, only a degree of score augmentation corresponding to increases in backbone sizes is provided. However, it can be seen that BMTPT achieves on-par-with performance compared to ATTEMPT and MPT, on BoolQ, MultiRC and WiC. In the other tasks within the GLUE/SuperGLUE benchmark, BMTPT generally surpasses both ATTEMPT and MPT when using T5-base. We believe this superiority pattern would be maintained when scaled up to T5-large or T5-3B.

---

### Official Review · Reviewer_AbeE · 2023-08-12

**Soundness:** 2

**Excitement:**

3: Ambivalent: It has merits (e.g., it reports state-of-the-art results, the idea is nice), but there are key weaknesses (e.g., it describes incremental work), and it can significantly benefit from another round of revision. However, I won't object to accepting it if my co-reviewers champion it.

**Missing References:**

P-Tuning: Prompt Tuning Can Be Comparable to Fine-tuning Across Scales and Tasks

**Paper Topic And Main Contributions:**

This paper solves the problem of  multi-task transfer learning for soft prompt tuning, considers the correlation among source tasks for better transfer to target tasks, and propose a Bayesian approach where they obtain representative source prompts corresponding to the samples from the posterior utilizing Stein Variational Gradient Descent (SVGD). Extensive experimental results on the standard benchmark NLP tasks show the advantage of the proposed Bayesian multi-task transfer learning approach. Besides, it has a high degree of parameter-efficient.

**Reasons To Accept:**

The proposed method demonstrates commendable innovation, standing apart from mere amalgamation of existing models. The novel approach showcases tangible efficacy without introducing additional parameters.

**Reasons To Reject:**

1.The absence of a computational complexity analysis, coupled with marginal and non-significant experimental performance improvements, raises concerns regarding the practical significance. If the proposed model introduces high computational complexity and yields only marginal gains, its real-world utility may be limited.

2.The paper lacks comparative analysis with some important baselines, such as P-tuning v2.

3. Additionally, the theoretical substantiation for the proposed method is insufficiently detailed.

4. The exclusive use of a single language model, T5-base, as the backbone prompts doubts about the general applicability of the proposed approach across a broader spectrum of models.

5. Furthermore, the paper lacks visual or interpretable analyses that incorporate concrete natural language statements.

Considering these points, I respectfully recommend that the authors thoroughly address these shortcomings to enhance the paper's overall quality and potential for contribution before reconsidering it for acceptance.

**Reproducibility:**

3: Could reproduce the results with some difficulty. The settings of parameters are underspecified or subjectively determined; the training/evaluation data are not widely available.

**Reviewer Confidence:**

3: Pretty sure, but there's a chance I missed something. Although I have a good feel for this area in general, I did not carefully check the paper's details, e.g., the math, experimental design, or novelty.

---

> ### Author Rebuttal · Authors · 2023-08-29
>
> Thank you for your detailed review and for acknowledging the innovation and efficiency of our work. We are gratified that you recognize the value of our approach, which uses Stein Variational Gradient Descent (SVGD) to obtain representative source prompts. Our method aims for both effectiveness and parameter efficiency, without requiring any additional mechanisms.
>
> ### **Computational Complexity Analyses**
> The additional computational requirements for BMTPT are primarily due to the calculation of Damped SVGD updates during source posterior learning and the computation of regularization terms originating from the prior during target adaptation. Given that Damped SVGD primarily involves computing RBF kernel values among a limited number of particles (in our case, five particles), the computational cost is minimal. Likewise, the regularization calculation from the log-likelihood of the prior during target adaptation is also negligible. As a result, BMTPT can learn the prompt posterior from source tasks without imposing a significant computational burden.
>
> This justifies the use of BMTPT for soft prompt tuning, as it delivers high performance with minimal computational demands. As evidenced by the tables in our initial response to reviewer qA2O, BMTPT exhibits superior performance across a large portion of the GLUE/SuperGLUE datasets. Furthermore, we have also discovered that our method shows advantages in the few-shot setting, indicating greater efficiency in real-world scenarios. Please refer to the table in the initial response to reviewer *g8bk*.
>
>
> ### **Theoretical Framework of the Proposed Method**
> We appreciate the reviewer's inquiry about the theoretical substantiation for our methodology. Our approach is fundamentally rooted in Bayesian inference, a paradigm which naturally allows us to update our understanding of uncertainty based on the data from the source task. Within this Bayesian framework, conventional variational inference techniques have been shown to fall short in capturing complex, multi-modal distributions.
>
> In this regard, we employ Stein Variational Gradient Descent (SVGD). Unlike traditional methods, SVGD is a non-parametric approach whose particles can represent flexible class of approximating distributions, which enables it to model intricate distributions with higher fidelity.
>
> ### **Choice of Backbone Model**
> We chose to use T5-{base, large, 3B} as backbone for fair comparison with notable baselines such as PT, SPoT, ATTEMPT and MPT, because these baselines adopted T5 as backbone in their experiments.
>
> However, to address the reviewer's concern, we are currently running experiment using BERT-large as backbone, to showcase that BMTPT is effective on LMs other than T5. We will report the performance of BERT-large based BMTPT and compare it with regular prompt tuning and **P-tuning V2** (used BERT-large as backbone) in the discussion period.
>
> ### **Need for Visual or Interpretable Analysis Incorporating Concrete Natural Language Statements**
> We will subsequently include empirical results for natural language generation tasks in the discussion period. Should you have specific experiments in mind, we would welcome your recommendations.

---

### Official Review · Reviewer_qA2o · 2023-08-15

**Soundness:** 3

**Excitement:**

3: Ambivalent: It has merits (e.g., it reports state-of-the-art results, the idea is nice), but there are key weaknesses (e.g., it describes incremental work), and it can significantly benefit from another round of revision. However, I won't object to accepting it if my co-reviewers champion it.

**Paper Topic And Main Contributions:**

This paper is about Bayesian Multi-Task Transfer Learning for Soft Prompt Tuning. The main problem addressed in this paper is how to improve the performance of pre-trained language models on target tasks by fine-tuning them with task-specific prompts. The paper proposes a novel approach to prompt tuning that takes into account the correlation among source tasks for better transfer to target tasks. The technical approaches, Partical-based Vartiinal Inference and Stein Variational Gradient Descent, are applied to solve the multi-task prompt tuning problem. Specifically, the proposed approach aims to learn a global posterior approximation of source tasks such that it can be transferred to downstream target tasks easily and efficiently.

**Reasons To Accept:**

1. A Bayesian multi-task transfer learning approach to prompt tuning that outperforms state-of-the-art methods in many settings.
2. A soft prompt tuning method that allows for more flexible and efficient transfer learning by using a continuous relaxation of the prompt vectors.
3. An extensive experimental evaluation on standard benchmark NLP tasks that demonstrate the effectiveness of the proposed approach and its superiority over existing methods.

**Reasons To Reject:**

1. Technically, the applied approaches are not novel techniques and the main novelty is applying them to solve a multitask learning problem. Multi-task prompt tuning is only one specific instance of multi-task learning.
2. Regardless of learning the posterior distribution of source tasks, the practical implementation seems to be kind of similar to the mentioned reference, MPT, esp., in terms of prompt decomposition, parameter efficiency, and the source training/target adaptation process. The authors might need to explain or clarify clearly the fundamental differences with MPT in the paper.
3. The empirical results are marginal or nearly the same as the baselines. The only vast improvement comes from the CoLA task in GLUE, improving it from 60+ in baseline to 86. The rest of all tasks are nearly the same or very marginal improvement over ATTEMPT or MPT. Therefore, the empirical improvement isn't impressive.

**Reproducibility:**

4: Could mostly reproduce the results, but there may be some variation because of sample variance or minor variations in their interpretation of the protocol or method.

**Reviewer Confidence:**

5: Positive that my evaluation is correct. I read the paper very carefully and I am very familiar with related work.

---

> ### Author Rebuttal · Authors · 2023-08-29
>
> Thank you for your insightful review and acknowledgment of our work's contributions. We are pleased that our approach has been noted for its comparative advantages in various environments. The effectiveness of our method, enabled by optimizing prompt vectors through a Bayesian approach, aims to offer a more effective approach to transfer learning.
>
> ### **Contrast with Traditional Multi-Task Learning**
> We agree that our methodology seems similar with typical multi-task learning, in terms of the use of multi-source data in utilizing SVGD with prompt particles. However, this is largely a byproduct of our Bayesian formulation. Our aim is to create the prompt with more robust representation of general knowledge by leveraging posterior information via Bayesian approach. In doing so, we utilize SVGD to update the particles (each particle is an instantiation of a prompt) to approximate the joint posterior distribution from source domain datasets. This unique formulation and its implementation differentiate our work from existing multi-task learning paradigms in the context of prompt tuning.
>
> ### **Comparison with MPT**
> Our work bears resemblance to MPT in how we construct a soft prompt by composing full-rank and low-rank matrices during the target adaptation stage. However, it is important to understand that both our motivation and our detailed approach to composing full-rank and low-rank matrices differ significantly from the strategies employed by MPT.
>
> MPT uses prompt decomposition to separate shared and task-specific information during source task training. In contrast, BMTPT employs a similar decomposition but applies it to the target task(s) during the task adaptation stage. We create a full-rank matrix by aggregating SVGD particles that have learned the source posterior and combine it with a low-rank matrix through element-wise multiplication. The low-rank matrix is introduced per target task, and it is designed to handle multi-target task adaptation scenarios.
>
> ### **Empirical Performance**
> As the reviewer noted, our empirical results may not explicitly surpass on all of the existing NLP tasks. However, we would like to emphasize that our method demonstrates superior performance across a significant portion of the GLUE/SuperGLUE datasets. Moreover, it achieves this without requiring any components beyond the soft prompts and the backbone model, unlike other baselines, which do require additional elements (e.g., ATTEMPT incorporates an attention module, MPT utilizes a separate backbone model for distillation).
>
> Additionally, we re-present the tables for the experiments include the number of tunable parameters for comparison. Our BMTPT demonstrates superior performance compared to baselines that utilize a larger number of parameters or additional networks.
>
> In the tables, it is evident that other baselines such as Adapters (1.9M params), BitFit (280K params) and ATTEMPT (232K params) utilize considerably more parameters compared to our method, BMTPT (77.6K params). Despite this, we demonstrate performance that is largely superior, or at the very least, comparable, across both single and multiple tasks within the GLUE/SuperGLUE datasets.
>
> ### Result on GLUE Datasets (Single Target Task)
> | Method        | #params  | MNLI  | QQP  | QNLI | SST-2 | STS-B | MRPC | RTE  | CoLA | Avg. |
> |---------------:|:---------:|:-------:|:------:|:------:|:------:|:------:|:------:|:------:|:------:|:------:|
> | Adapters      | 1.9M    | **86.5**  | 90.2 | 93.2 | 93.8  | 90.7  | 85.3 | 71.9 | 64.0 | 84.5 |
> | BitFit        | 280K    | 85.3  | 90.1 | 93.0 | 94.2  | **90.9**  | 86.8 | 67.6 | 58.2 | 83.3 |
> | PT            | 76.8K   | 81.3  | 89.7 | 92.8 | 90.9  | 89.5  | 68.1 | 54.7 | 10.6 | 72.2 |
> | Vanilla transfer PT | 76.8K  | 85.8  | 86.9 | 93.2 | 92.9  | 90.5  | 87.1 | 77.0 | 83.2 | 87.1 |
> | SPoT          | 76.8K   | 85.4  | 90.1 | 93.0 | 93.4  | 90.0  | 79.7 | 69.8 | 57.1 | 82.3 |
> | ATTEMPT       | 232K    | 84.3  | **90.3** | 93.0 | 93.2  | 89.7  | 85.7 | 73.4 | 57.4 | 83.4 |
> | MPT          | 77.6K   | 85.9  | **90.3** | 93.1 | 93.8  | 90.4  | **89.1** | 79.4 | 62.4 | 85.6 |
> | BMTPT (Ours)  | 77.6K | 86.2  | **90.3** | **93.4** | **94.4**  | **90.9** | 87.2 | **81.3** | **86.6** | **88.7** |
>
> ### Result on GLUE Datasets (Multiple Target Tasks)
> | Method        | #params  | MNLI  | QQP  | QNLI | SST-2 | STS-B | MRPC | RTE  | CoLA | Avg. |
> |---------------:|:---------:|:-------:|:------:|:------:|:------:|:------:|:------:|:------:|:------:|:------:|
> | Adapters     | 1.9M    | **86.3**  | **90.5** | 93.2 | 93.0  | 89.9  | **90.2** | 70.3 | 61.5 | 84.4 |
> | HyperFormer  | 280K    | 85.7  | 90.0 | 93.0 | 94.0  | 89.7  | 87.2 | 75.4 | 63.7 | 84.8 |
> | HyperDecoder | 76.8K   | 86.0  | **90.5** | **93.4** | 94.0  | 90.5  | 87.7 | 71.7 | 55.9 | 83.7 |
> | ATTEMPT      | 232K    | 83.8  | 90.0 | 93.1 | 93.7  | 90.8  | 86.1 | 79.9 | 64.3 | 85.2 |
> | MPT          | 77.6K   | 84.3  | 90.0 | 93.0 | 93.3  | 90.4  | 89.2 | **82.7** | 63.5 | 85.8 |
> | BMTPT (Ours) | 77.6K | 85.9  | 90.2 | 93.2 | **95.3**  | **91.2**  | 86.9 | 80.9 | **85.6** | **88.7** |
>
> ### Result on SuperGLUE Datasets (Single Target Task)
> | Method              | #params  | Multirc  | BoolQ  | WiC | WSC | CB | Avg. |
> |---------------------|:-------:|:--------:|:------:|:---:|:---:|:--:|:-----:|
> | Adapters            | 1.9M    | **75.9**     | **82.5** | 67.1| **67.3** | **85.7**| **75.7**  |
> | BitFit              | 280K    | 74.5     | 79.6   | **70.0**| 59.6 | 78.6| 72.5  |
> | PT                  | 76.8K   | 58.7     | 61.7   | 48.9| 51.9 | 67.9| 57.8  |
> | Vanilla transfer PT | 76.8K   | 72.2     | 77.9   | 65.5| **67.3** | 78.6| 72.3  |
> | SPoT                | 76.8K   | 74.0     | 77.2   | 67.0| 50.0 | 46.4| 62.9  |
> | ATTEMPT             | 232K    | 74.4     | 78.8   | 66.8| 53.8 | 78.6| 70.5  |
> | MPT                 | 77.6K   | 74.8     | 79.6   | 69.0| **67.3** | 79.8| 74.1  |
> | BMTPT (Ours)        | 77.6K| 72.4 | 80.3| 67.4| **67.3**| **85.7**| 74.6 |
>
> ### Result on SuperGLUE Datasets (Multiple Target Tasks)
> | Method              | #params  | Multirc  | BoolQ  | WiC | WSC | CB | Avg.  |
> |---------------------|:-------:|:--------:|:------:|:---:|:---:|:--:|:-----:|
> | ATTEMPT            | 232K    | 74.4     | 78.3   | 66.5| **69.2** | 82.1| 74.1  |
> | MPT                | 77.6K   | **74.8**     | 79.2   | **70.2**| 67.3 | **89.3**| **76.1**  |
> | BMTPT (Ours)       | 77.6K | 72.3 | **80.1** | 67.7| 67.3 | **89.3**| 75.3 |
>
> Furthermore, we would like to direct the reviewer to the empirical performance in the few-shot experiment, as displayed in the initial response to reviewer *g8bk*. The result demonstrates that our prior guides the prompts toward an efficient initial point for task adaptation in low-resource scenario.

---

### Meta-Review · Program_Chairs · 2023-09-09

**Recommendation:** 3

**Metareview:**

The paper introduces BMTPT to address the challenge of improving the performance of PTLMs on target tasks through task-specific prompt fine-tuning. BMTPT takes into consideration the correlation among source tasks to enhance the transfer to target tasks. Particle-based Variational Inference and Stein Variational Gradient Descent (SVGD) are utilized to solve the multi-task prompt tuning problem. Specifically, BMTPT aims to learn a global posterior approximation for source tasks, facilitating efficient transfer to downstream target tasks. Experimental results on standard benchmark NLP tasks demonstrate the effectiveness and parameter efficiency of the proposed Bayesian multi-task transfer learning approach, outperforming existing methods by capturing task correlations and leveraging this knowledge to improve target task performance.


The majority of reviewers accept that the soundness of this work is moderate and the excitement level is decent. Reviewer AbeE has some concerns but the reviewers also provided substantive responses. However, the reviewer did not engage in discussions and I found responses by the authors at least relevant.

---

### Decision · Program_Chairs · 2023-10-07

**Decision:**

Accept-Findings

**Comment:**

The paper introduces BMTPT to address the challenge of improving the performance of PTLMs on target tasks through task-specific prompt fine-tuning. BMTPT takes into consideration the correlation among source tasks to enhance the transfer to target tasks. Particle-based Variational Inference and Stein Variational Gradient Descent (SVGD) are utilized to solve the multi-task prompt tuning problem. Specifically, BMTPT aims to learn a global posterior approximation for source tasks, facilitating efficient transfer to downstream target tasks. Experimental results on standard benchmark NLP tasks demonstrate the effectiveness and parameter efficiency of the proposed Bayesian multi-task transfer learning approach, outperforming existing methods by capturing task correlations and leveraging this knowledge to improve target task performance.


The majority of reviewers accept that the soundness of this work is moderate and the excitement level is decent. Reviewer AbeE has some concerns but the reviewers also provided substantive responses. However, the reviewer did not engage in discussions and I found responses by the authors at least relevant.